# Increased Monocytic Myeloid-Derived Suppressor Cells in Whole Blood Predict Poor Prognosis in Patients with Plasma Cell Myeloma

**DOI:** 10.3390/jcm10204717

**Published:** 2021-10-14

**Authors:** Mi-Hyun Bae, Chan-Jeoung Park, Cheolwon Suh

**Affiliations:** 1Department of Laboratory Medicine, Hanyang University Guri Hospital, Hanyang University College of Medicine, Guri 11923, Korea; mhbae@hanyang.ac.kr; 2Department of Laboratory Medicine, University of Ulsan College of Medicine and Asan Medical Center, Seoul 05505, Korea; 3Department of Internal Medicine, University of Ulsan College of Medicine and Asan Medical Center, Seoul 05505, Korea; csuh@amc.seoul.kr

**Keywords:** monocytic myeloid-derived suppressor cells, plasma cell myeloma, whole blood, prognosis, survival

## Abstract

Myeloid-derived suppressor cells (MDSCs) are heterogeneous populations of immature myeloid cells with immunosuppressive effects that have prognostic potential in patients with malignancies; however, survival analysis studies are sparse. In this study, the prognostic implication of MDSCs was investigated in peripheral blood (PB) and bone marrow (BM) samples from 81 patients with plasma cell myeloma at diagnosis. MDSCs were quantified as monocytic MDSCs (mMDSCs) (CD11b^+^HLA-DR^−/low^CD14^+^) and granulocytic MDSCs with neutrophils (gMDSCs-N) (CD11b^+^HLA-DR^−/low^CD14^−^CD33^+^CD15^+^). Serum creatinine and lactate dehydrogenase levels showed a moderate correlation with all MDSC types, except BM-gMDSCs-N; mMDSCs correlated with serum β2-microglobulin level, and PB-mMDSCs showed an inverse correlation with hemoglobin. PB-mMDSC levels were significantly higher in patients with progressive disease than those in patients at diagnosis and complete response. BM-mMDSC levels in patients with progressive disease were also higher than those in patients at diagnosis. Patients with high mMDSCs showed significantly poorer prognosis than patients with low mMDSCs. Multivariate analysis showed high PB-mMDSCs (≥0.3%) as a significant adverse prognostic marker for overall survival. This study demonstrated the independent adverse prognostic impact of PB-mMDSCs in patients with myeloma. PB-mMDSC measurement using whole blood is readily accessible in clinical laboratories, and may be used as a prognostic marker in clinical practice.

## 1. Introduction

Myeloid-derived suppressor cells (MDSCs) are heterogeneous populations of immature myeloid cells that have the ability to regulate immune responses [1]. Their numbers are elevated in chronic pathological conditions, such as infection, inflammation, and cancer [2,3]. Earlier studies on solid tumors reported that the accumulation of MDSCs promotes tumor growth via facilitating tumor escape from the host’s immune system and chemotherapy [2,4,5]; moreover, high MDSC counts in patients with cancer are associated with poor prognosis [6]. The tumor-promoting role of MDSCs suggests that they may be a potential target for immunomodulatory therapy [7].

Studies exploring the role of MDSCs in hematological malignancies, including plasma cell myeloma (PCM) [8,9], have reported that MDSC levels are increased in peripheral blood (PB) of patients with active PCM compared with those in healthy individuals, and the immune-suppressive and tumor-promoting effects of MDSCs in the PCM microenvironment have been confirmed [10,11]. As accumulation of neoplastic plasma cells as well as myelopoiesis to generate MDSCs occur in the bone marrow (BM), clinical investigations are important for identifying the real clinical impact of MDSC accumulation in patients with PCM. However, clinical studies based on MDSCs in PCM are sparse, and so far, only one study has reported the adverse prognostic impact of monocytic MDSCs (mMDSCs) in the PB of patients with PCM at the pre-autologous stem cell transplantation phase [12].

Flow cytometry is an essential tool for identifying the two major MDSC subtypes in humans: granulocytic MDSCs (gMDSCs, CD11b^+^HLA-DR^−/low^CD14^−^CD33^+^CD15^+^) and mMDSCs (CD11b^+^HLA-DR^−/low^CD14^+^) [13]. Traditionally, studies on MDSCs are performed with peripheral blood mononuclear cells (PBMCs), which require an additional cell separation step from whole blood (WB); however, PBMC preparation is a laborious procedure in clinical practice, and it possesses potential technical variations [14]. For this reason, MDSC phenotyping using WB was recently introduced to unveil the prognostic impact of MDSCs in patients with malignancies [15]. Because there is no single marker to discriminate neutrophils from MDSCs at present, gMDSC population gated from WB or whole BM could include mature neutrophils. Thus, the population is designated as gMDSCs-N (gMDSCs with neutrophils) in this study. Here, gMDSCs-N and mMDSCs were quantified from WB and BM in patients with PCM at diagnosis using flow cytometry, and the prognostic impact was investigated to extend the prognostic implication of MDSCs in the clinical management of PCM.

## 2. Materials and Methods

### 2.1. Patients

A total of 97 consecutive adult patients who were newly diagnosed with PCM at Asan Medical Center (Seoul, Korea) between August 2013 and September 2014 were included in this study. Patients were diagnosed according to the International Myeloma Working Group criteria [16]. PCM was defined as the presence of more than 10% of clonal BM plasma cells and signs of end organ damage (hypercalcemia, renal failure, anemia, or bone lesions). Cytogenetic analysis was performed using conventional G-banding and fluorescence in situ hybridization (FISH) for t(11;14), t(4;14), t(14;16), and del(17p). A total of 81 patients were eligible for myeloma therapy. A total of 33 patients under the age of 65 years were initially treated with a thalidomide plus dexamethasone regimen, and 25 patients among them received autologous hematopoietic cell transplantation (ASCT). Patients 65 years and older were treated with a combination regimen of bortezomib, melphalan, and prednisone, except two patients who received carfilzomib. Of the 97 patients with PCM, baseline parameters were evaluated and clinical outcome analysis was performed in 81 patients who received myeloma therapy.

MDSCs were also measured in 12 patients with PCM who were in complete response (CR, *n* = 6) or who had progressive disease (PD, *n* = 6) to compare MDSC level with disease status.

This study was approved by the Institutional Review Board of the Asan Medical Center (2016-1299); the subjects’ informed consent to participate in research was waived. This study included 72 patients who were enrolled in a previous study on circulating plasma cells [17].

### 2.2. Flow Cytometry

WB and BM samples were collected in tubes containing ethylenediaminetetraacetic acid (EDTA) from 97 patients with PCM at the time of diagnosis, 6 patients in CR, and 6 patients with PD. Samples were evaluated by performing flow cytometry within 24 h of collection. An amount of 100 uL of samples was labeled with specific antibodies for 20 min in the dark at room temperature, and erythrocytes were lysed using lysing solution for 10 min and washed and suspended in phosphate-buffered solution. Fluorescein isothiocyanate (FITC)-conjugated anti-human leukocyte antigen (HLA)-DR antibody, phycoerythrin (PE)-conjugated anti-CD11b antibody, peridinin-chlorophyll protein (PerCP)-conjugated anti-CD33 antibody, allophycocyanin (APC)-conjugated anti-CD15 antibody, and PE-Cy7-conjugated anti-CD14 antibody were simultaneously used to label samples (3–4 uL of each antibody). All antibodies were obtained from BD Biosciences (San Jose, CA, USA). Five-color flow cytometry was performed, and data were analyzed using FACSCanto II (BD Biosciences) and FACSDiva software (BD Biosciences). Fifty-thousand nucleated cells were acquired per tube and 50 events were assigned as the lower limit of detection. Immunophenotypes of mMDSCs (CD11b^+^HLA-DR^−/low^CD14^+^) and gMDSCs-N (CD11b^+^HLA-DR^−/low^CD14^−^CD33^+^CD15^+^) were determined (Figure 1). MDSCs were quantified as the proportion of nucleated cells analyzed, and absolute counts of PB MDSCs were calculated using white blood cell (WBC) counts of PB.

### 2.3. Statistical Analysis

All statistical analyses were performed using the Statistical Package for the Social Sciences (ver. 18.0; Chicago, IL, USA). Spearman’s rank correlation coefficient (***ρ***) and Mann–Whitney U test were used to analyze continuous variables, and the chi-square test was used for tabular data. The cutoff values for MDSC types were determined from receiver operating characteristic (ROC) curves. The Kaplan–Meier method with a log-rank test was used to evaluate overall survival (OS; time from study entry until death). The OS hazard ratios were estimated using Cox regression modeling. All tests were two-sided, and results with *p* < 0.05 were considered statistically significant.

## 3. Results

### 3.1. Patient Characteristics

Among the newly diagnosed 97 patients with PCM, 51 patients (52.6%) were female, and the median age of all patients was 66.5 years (37–87 years) (Table 1). The median follow-up duration was 62.2 months, and the mortality rate during the study was 57.7% (56/97). Cytogenetic analysis with G-banding was performed for all the patients, while FISH was performed for 61 patients (62.9%). Conventional cytogenetic analysis showed that 34 patients (35.1%) had monosomy 13, and three patients (3.1%) presented hypodiploidy with structural aberrations. One patient with t(11:14) was detected by conventional cytogenetics, and FISH analysis was not performed for him. Among the 61 patients for whom FISH was performed, t (11;14) was detected in 12 patients (19.7%), and t (4;14) was detected in 8 patients (13.1%). One patient showed 17p deletion, but t (14;16) was not observed. Using the risk stratification with International Staging System (ISS), 15 patients (15.5%) were stage I, 45 (46.4%) were stage II, and 37 (38.1%) were stage III. Patients in CR (*n* = 6) and PD (*n* = 6) were included in this study (Table 1). The data shown in the table were acquired through diagnostic workup except age. Patients’ ages were recorded at MDSC measurement.

Histologically, all patients newly diagnosed had plasmacytic type myeloma, except seven patients who showed plasmablastic type. Serum immunoglobulin (Ig) restriction was common for IgG (57, 58.8%), followed by light chain (13, 13.4%), IgA (12, 12.4%), and IgD (6, 6.2%). Nine patients did not show immunoglobulin restriction in serum, comprising four patients with Bence Jones proteinuria and five patients with non-secretory myeloma (data not shown).

### 3.2. Laboratory and Clinical Association with Levels of MDSCs

The median proportions of PB mMDSCs and gMDSCs-N quantified from 97 patients with PCM at diagnosis were 0.1% and 22.8%, respectively (Table 2). Median absolute numbers of PB MDSCs were 7.7 × 10^6^/L and 1128 × 10^6^/L for mMDSCs and gMDSCs-N, respectively. The median proportions of BM mMDSCs and gMDSCs-N were 0.2% and 24.9%, respectively. Spearman’s correlation analysis was performed to find the association between the levels of MDSCs and laboratory prognostic markers (Table 2). Serum creatinine and lactate dehydrogenase (LD) levels showed a moderate correlation with all types of MDSCs, except BM-gMDSCs-N (*ρ* values between 0.234 and 0.441, *p* values between < 0.001 and 0.021). PB and BM mMDSC levels, but not of gMDSCs-N, correlated with serum β2-microglobulin (β2M) levels (*ρ* = 0.353, 0.355, and 0.217, in order of PB%, PB number, and BM%, respectively, *p* < 0.001 for PB and *p* = 0.033 for BM). The proportions and numbers of PB-mMDSCs showed a significant inverse correlation with hemoglobin (*ρ* = −0.316 and −0.239, respectively, *p* = 0.002 and 0.018). BM cellularity and BM plasma cell counts were inversely correlated with BM-gMDSCs-N (*ρ* = −0.265 and −0.303, respectively, *p* = 0.009 and 0.003). Representative scatter plots with significant correlations are presented in Figure 2.

The newly diagnosed 97 patients were divided into two groups for each MDSC type (%) using cutoff values determined by ROC analysis. Table 3 shows the comparison of laboratory and clinical findings between the two groups divided by PB-mMDSCs (%) with a cutoff value of 0.3%. All measurements related to MDSCs were higher in the PB-mMDSCs ≥ 0.3% group (*n* = 28) than those in the PB-mMDSCs < 0.3% group (*n* = 69) (*p* < 0.001), except BM-gMDSCs-N (%), which were comparable between the two groups. In the PB-mMDSCs ≥ 0.3% group, creatinine, LD, β2M, and WBCs were higher (*p* < 0.001, *p* = 0.001, *p* < 0.001, and *p* = 0.007, respectively), and albumin and hemoglobin levels were lower (*p* = 0.018 and *p* < 0.001) than those in the PB-mMDSCs < 0.3% group. A total of 15 ISS stage I patients belonged to the PB-mMDSCs < 0.3% group, while there was no ISS I patient in the PB-mMDSCs ≥ 0.3% group. The proportions of ISS stage III patients and mortality were higher in the PB-mMDSCs ≥ 0.3% group than in the PB-mMDSCs < 0.3% group (*p* = 0.001 and 0.002, respectively). The proportion of high-risk cytogenetic features was not statistically different between the two groups.

The relationship between MDSC levels and the disease status was assessed by the Mann–Whitney U test, which analyzed each combination of two groups (diagnosis vs. CR, diagnosis vs. PD, and CR vs. PD). PB-mMDSC levels were significantly higher in patients with PD (median 1.2%) than those in patients at diagnosis (median 0.1%) and patients in CR (median 0.15%) (*p* = 0.003 and 0.026, respectively) (Figure 3a). PB-mMDSCs in patients in CR were not statistically different from those in patients at diagnosis. BM-mMDSCs proportion was higher in patients with PD (median 0.75%) than that in patients at diagnosis (median 0.2%, *p* = 0.007) (Figure 3b). However, gMDSCs-N relation to disease status did not show any significant differences between the groups (Figure 3c,d).

### 3.3. Prognostic Impact of MDSCs

Survival analysis was performed for 81 newly diagnosed patients who received myeloma treatment. The cutoff values determined by ROC analysis for PB and BM were 0.3% and 0.5% for mMDSCs (%) and 24.4% and 22.4% for gMDSCs-N (%), respectively (Figure 4). According to the Kaplan–Meier survival curve analysis, patients with PB-mMDSCs ≥ 0.3% and BM-mMDSCs ≥ 0.5% showed worse OS than the patients with PB-mMDSCs < 0.3% and BM-mMDSCs < 0.5%, respectively (*p* < 0.001 and *p* = 0.003) (Figure 5a,b). PB-gMDSCs-N ≥ 24.4% showed a trend for worse OS compared with that of PB-gMDSCs-N < 24.4%, but it was not significant (*p* = 0.144) (Figure 5c). BM-gMDSCs-N ≥ 22.4% was associated with better OS than BM-gMDSCs-N < 22.4% (*p* = 0.026), but the survival curves of the two groups overlapped until the mid-point of observation time (35 months from diagnosis) (Figure 5d).

In univariate Cox regression analysis for OS, MDSC types were primarily analyzed as continuous variables (Table 4). The increase in any mMDSC types was associated with a significant adverse prognosis (*p* < 0.001 for PB-mMDSCs and *p* = 0.006 for BM-mMDSCs). No gMDSCs-N types showed a prognostic impact. Patients at ISS stage III were associated with a poor OS compared with patients at stage I (*p* = 0.009). High-risk cytogenetics also showed poor OS (*p* = 0.042). Among the serum markers, LD ≥ 250 IU/L, β2M ≥ 5.5 μg/mL, and albumin < 3.5 g/dL were associated with shorter OS (*p* = 0.003, 0.009, and 0.037, respectively); low hemoglobin level < 8.5 g/L was associated with a trend of worse OS (*p* = 0.081). Increased BM plasma cells and BM cellularity were associated with poor OS (*p* = 0.017 and 0.006, respectively). In multivariate analysis, age ≥ 65 years, ISS stage, high risk cytogenetic features, and known PCM laboratory prognostic markers (LD and hemoglobin) were considered covariates. Albumin and β2M were not included in multivariate analysis because they were factors of ISS. Respective PB and BM MDSCs, including mMDSCs and gMDSCs-N, were analyzed as dichotomous variables in the multivariate analysis. Only a level of PB-mMDSCs ≥ 0.3% retained its significance in the multivariate analysis for OS (*p* = 0.040).

## 4. Discussion

MDSCs are immature myeloid cells that have defective differentiation, that contribute to tumor immune tolerance, and that promote tumor progression in several malignancies [5,8,13]. MDSCs suppress anti-tumor T cell responses through arginine depletion, production of nitric oxide and reactive oxygen species, and regulatory T cell development [13]. MDSCs induce myeloma cell growth, and myeloma cell-induced MDSCs develop bidirectionally in the BM microenvironment [10]. Proportions of gMDSC (CD11b^+^HLA-DR^-/low^CD14^−^CD33^+^CD15^+^) are higher in patients with progressive [10] or refractory PCM than those in healthy controls [18]. One study showed that patients with stable disease (including CR) showed increased PB-gMDSC levels compared with those of healthy controls [18], whereas another study suggested that BM-gMDSC levels in patients at diagnosis or relapse were not different from PB-gMDSC levels in healthy controls, but only relapsed/refractory state was associated with higher MDSC level [10]. In another study that analyzed mMDSCs (CD14^+^HLA-DR^−/low^ cells) [11], PB- and BM-mMDSC levels from newly diagnosed or relapsed PCM patients were higher than MDSC levels from the patients at remission, and PB-mMDSC levels from patients at remission were similarly low with MDSC levels of healthy donors.

PB-mMDSC levels were higher in patients with progressive PCM than those at diagnosis or in CR in the present study. None of the previous studies have reported a high proportion of mMDSCs in patients with progressive PCM. Furthermore, BM-mMDSC levels were also higher in patients with PD than at diagnosis. However, no significant differences were noted in mMDSC levels between patients at diagnosis and those in CR in this study, unlike the previous study [11]. This may be due to low number of patients in CR (*n* = 6) or different MDSC measurement methods (WB in this study vs. mononuclear cell separation) or different calculation models where MDSC levels were estimated as percentages among CD14^+^ monocytes in the previous report [11], rather than proportion among the total examined cells in this study. Since the flow cytometric immunophenotyping strategies and calculation models used are heterogeneous among different studies, it is hard to directly compare MDSC levels with previous studies. Hence, further experiments are needed to verify the outcomes of the present study and whether the PB-mMDSC levels in the CR state and healthy donors are comparable with those using WB MDSC measurements. Additionally, the standardization of methods would help broaden the application of techniques, such as flow cytometric quantification of MDSCs using WB.

MDSCs have been the focus of many studies because they are a candidate prognostic marker and an immunotherapy target for treating malignancies. As previous studies mainly focused on clarifying the MDSC immune suppressive function, they used blood samples processed with the mononuclear cell separation [1]. However, as the human immune system is highly complex and tightly regulated by interactions between immune cells, fractionated PBMCs may distort immunologic patterns and amplify technical variations. Thus, using WB in human immunological studies has been proposed to minimize technical artifacts and provide more robust and representative in vivo immunological state results [14]. The most notable difference between PBMCs and WB in immune cell quantification is the removal of neutrophils after cell separation. Because there is no single marker to discriminate neutrophils from MDSCs at present, development of a novel marker will be helpful for continuing WB MDSCs studies. In addition, as BM neutrophils are functionally similar to gMDSCs in protection of myeloma cells from chemotherapy, and as the protective effect was mediated by soluble factors from neutrophils and gMDSCs [19], the investigation to reveal the functional and prognostic role of the gMDSCs population gated from WB, which includes neutrophils, would provide more conclusive inferences in human immunology.

The accumulation of clonal plasma cells in BM is a particular characteristic of PCM and one of the most distinguishable features between myeloma and solid tumors in MDSC studies. As MDSCs originate from BM cells through myelopoiesis, myeloma cells and MDSCs coexist in the BM and share the microenvironment. The bidirectional interaction between myeloma cells and MDSCs for promoting the proliferation of each other has been demonstrated in previous cellular studies [10], but the relationship in vivo has not been studied. In this study, there were significant inverse correlations between BM-gMDSCs-N levels and BM cellularity as well as plasma cells, which may be due to occupying the same BM space and highly proliferating clonal plasma cells disturbing MDSC myelopoiesis, including neutrophils. Here, patients with high BM-gMDSCs-N ≥ 22.4% showed better OS than that of patients with BM-gMDSCs-N < 22.4% based on Kaplan–Meier survival analysis. However, both survival curves overlapped until 35-month follow-up, and the prognostic impact was not significant in multivariate analysis (data not shown). Thus, the observed prognostic impact of the high BM-gMDSCs-N group may be a result of the compounding effects of lower tumor burden and increased myelopoiesis rather than a direct role of BM-gMDSCs at initial diagnosis.

MDSCs in WB and BM aspirates were quantified and analyzed using flow cytometry in patients with myeloma in the present study. This study demonstrated that an increased level of PB-mMDSCs at diagnosis was an independent poor prognostic factor in patients with PCM, which has not been previously reported. PB-mMDSCs detected in pre-ASCT were correlated with poor clinical outcome after ASCT in patients with PCM in a previous study using PBMCs [12]. Almost all previous studies used PBMCs to detect MDSCs, but PBMC preparation is laborious, and the process is rarely applied in clinical laboratories. Despite the massive data on the role of MDSCs in malignancies, MDSC isolation is one of the main obstacles in adopting MDSC measurement methods in clinical practice. The PB-mMDSC measurement method used in this study can be easily applied in real clinical practice because clinical laboratories are skillful at flow cytometric measurement with WB samples.

This study had limitations. The first limitation was that patients were treated with different regimens due to the National Health Insurance policy in Korea. Bortezomib was not allowed for initial PCM therapy in patients aged < 65 years during the study period. These patients were treated initially with thalidomide plus dexamethasone and followed with ASCT. Patients aged 65 years or above were allowed to start treatment with a combination regimen of bortezomib, melphalan, and prednisone. In spite of the different treatment regimens, the adverse prognostic impact of PB-mMDSCs on OS was independently significant based on the multivariate analysis. Further studies with equivalent treatment protocols, including currently used novel agents, are needed to confirm these results. The second limitation was that we were unable to conduct functional studies with MDSCs. This study is the first attempt at using WB in patients with malignancy, and a large scale prospective study began in 2020 in Europe [15]. Thus, demonstration of MDSC functional status in WB is anticipated through continuing studies using WB.

In conclusion, the flow cytometric quantification of MDSCs using WB and BM aspirates demonstrated that PB and BM mMDSC counts were increased in patients with progressive PCM compared with those in patients at diagnosis and in CR and that PB-mMDSCs ≥ 0.3% at diagnosis had an independent adverse prognostic impact on OS. The flow cytometric PB-mMDSC measurement using WB is readily accessible from clinical laboratories, and PB-mMDSCs may be used as a prognostic marker for PCM in clinical practice in the near future.

## Figures and Tables

**Figure 1 jcm-10-04717-f001:**
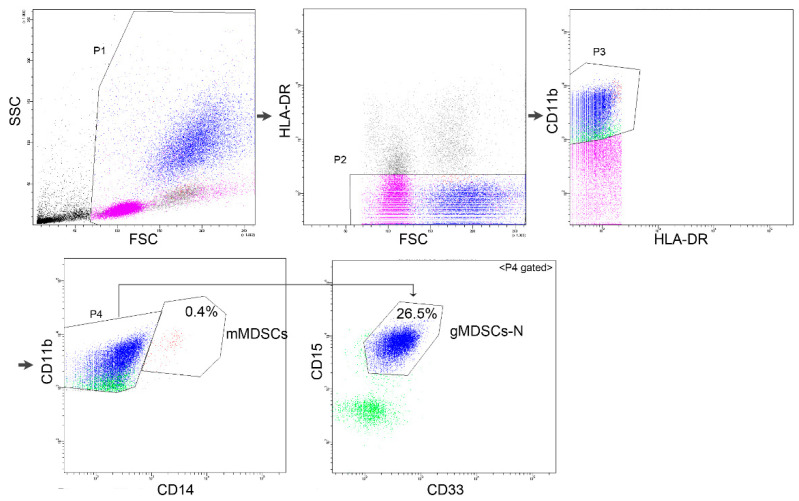
Gating strategy for myeloid-derived suppressor cells in peripheral blood. Monocytic MDSCs are defined as leukocytes of CD11b^+^HLA-DR^−/low^CD14^+^ immunophenotype and granulocytic MDSCs-N as CD11b^+^HLA-DR^−/low^CD14^−^CD33^+^CD15^+^ immunophenotype. Abbreviations: SSC, side scatter; FSC, forward scatter; HLA-DR, human leukocyte antigen-DR; mMDSCs, monocytic myeloid-derived suppressor cells; gMDSCs-N, granulocytic myeloid-derived suppressor cells with neutrophils.

**Figure 2 jcm-10-04717-f002:**
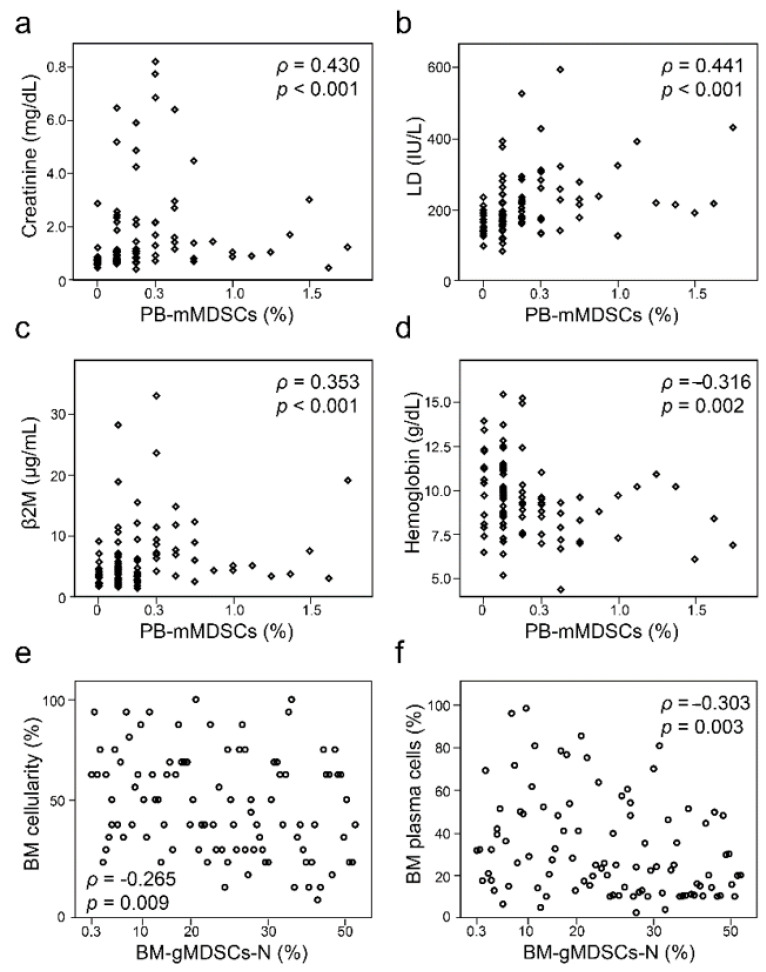
Scatter plots with significant correlation between laboratory measurements and myeloid-derived suppressor cells of 97 patients with plasma cell myeloma. Peripheral blood monocytic MDSCs show correlation with serum creatinine, lactate dehydrogenase, and β2-microglobulin (**a**–**c**) and inverse correlation with hemoglobin level (**d**). Bone marrow granulocytic MDSCs with neutrophils inversely correlate with bone marrow cellularity and plasma cells (**e**,**f**). Abbreviations: Cr, creatinine; LD, lactate dehydrogenase; β2M, β2-microglobulin; PB-mMDSCs, peripheral blood monocytic myeloid-derived suppressor cells; BM-gMDSCs-N, bone marrow granulocytic myeloid-derived suppressor cells with neutrophils.

**Figure 3 jcm-10-04717-f003:**
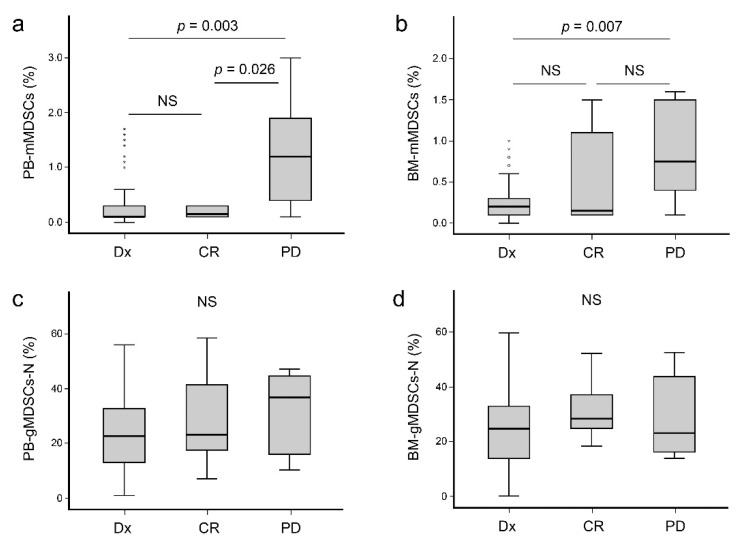
Comparison of myeloid-derived suppressor cells related to disease status of plasma cell myeloma. (**a**) PB-mMDSCs of patients with progressive disease showed higher levels than those of patients at diagnosis and in complete response (*p* = 0.003 and 0.026, respectively). (**b**) BM-mMDSC levels were higher in patients with progressive disease than those in patients at diagnosis (*p* = 0.007). (**c**,**d**) No gMDSCs-N types showed statistical differences between disease states. Abbreviations: Dx, diagnosis; CR, complete response; PD, progressive disease; PB-mMDSCs, peripheral blood monocytic myeloid-derived suppressor cells; BM-mMDSCs, bone marrow monocytic myeloid-derived suppressor cells; PB-gMDSCs-N, peripheral blood granulocytic myeloid-derived suppressor cells with neutrophils; BM-gMDSCs-N, bone marrow granulocytic myeloid-derived suppressor cells with neutrophils.

**Figure 4 jcm-10-04717-f004:**
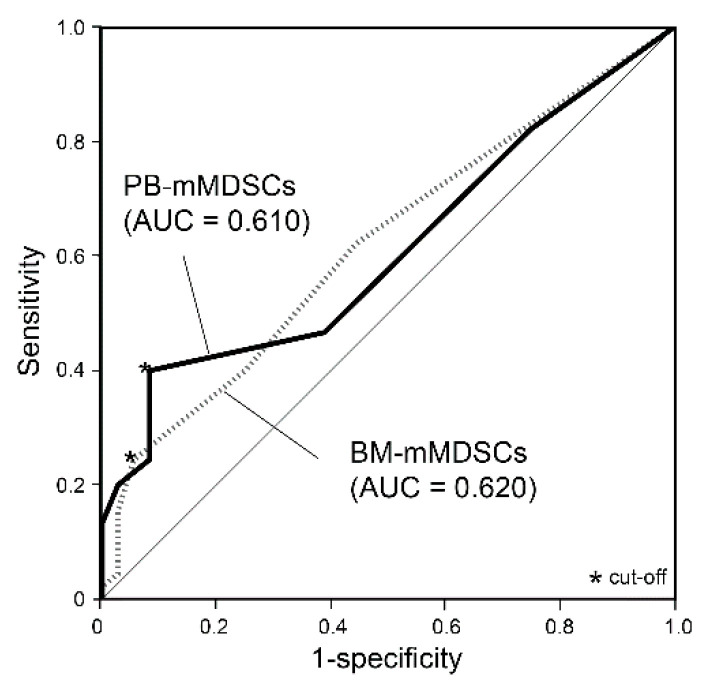
Receiver operating characteristic curves to determine cutoff values of monocytic myeloid-derived suppressor cells for mortality. Abbreviations: PB; peripheral blood, BM; bone marrow, mMDSCs; monocytic myeloid-derived suppressor cells.

**Figure 5 jcm-10-04717-f005:**
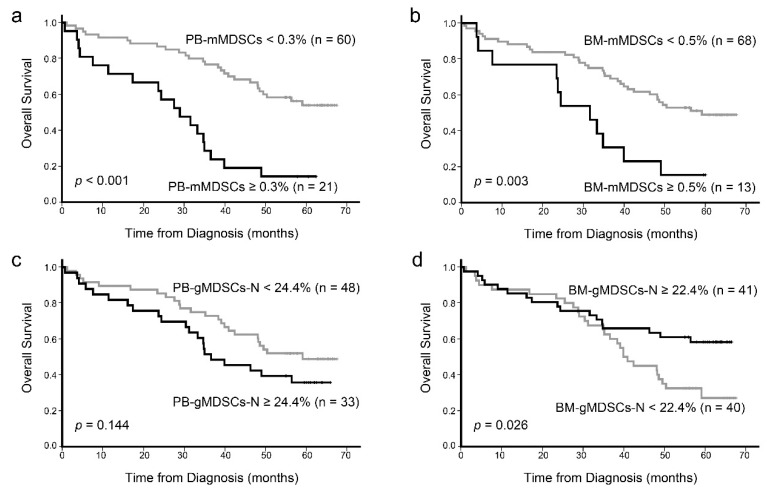
Kaplan–Meier survival curves for overall survival in 81 patients with plasma cell myeloma in which patient groups were categorized based on the proportion of four types of myeloid-derived suppressor cells measured at diagnosis. Patients with PB-mMDSCs ≥ 0.3% and BM-mMDSCs ≥ 0.5% showed worse OS than the patients with PB-mMDSCs < 0.3% and BM-mMDSCs < 0.5%, respectively (*p* < 0.001 and *p* = 0.003) (**a**,**b**). PB-gMDSCs-N ≥ 24.4% showed a trend for worse OS compared to that of PB-gMDSCs-N < 24.4%, but it was not significant (p = 0.144) (**c**). BM-gMDSCs-N ≥ 22.4% was associated with better OS than BM-gMDSCs-N < 22.4% (*p* = 0.026), but the survival curves of the two groups overlapped until the mid-point of observation time (35 months from diagnosis) (**d**).Abbreviations: OS, Overall survival; PB-mMDSCs, peripheral blood monocytic myeloid-derived suppressor cells; BM-mMDSCs, bone marrow monocytic myeloid-derived suppressor cells; PB-gMDSCs-N, peripheral blood granulocytic myeloid-derived suppressor cells with neutrophils; BM-gMDSCs-N, bone marrow granulocytic myeloid-derived suppressor cells with neutrophils.

**Table 1 jcm-10-04717-t001:** Clinical and cytogenetic findings of 109 patients with plasma cell myeloma.

Characteristics	Disease Status		
	Diagnosis	CR	PD
Total patients, *n*	97	6	6
Sex, *n* (%)			
Male	46 (47.4)	2 (33.3)	4 (66.7)
Female	51 (52.6)	4 (66.7)	2 (33.3)
Median age (range), years	66.5 (37–87)	60 (44–71) **	59.5 (39–63) **
Median follow-up duration (range), months	62.2 (54.5–67.6)	–	–
Total death, *n* (%)	56 (57.7)	–	–
Conventional cytogenetics abnormalities, *n* (%)			
Available patients	97 (100)	5 (83.3)	6 (100)
Monosomy 13	34 (35.1)	1 (20.0)	2 (33.3)
Hypodiploidy with structural aberrations	3 (3.1)	0 (0.0)	0 (0.0)
t(11;14) *	1 (1.0)	0 (0.0)	1 (16.7)
FISH abnormalities, *n* (%)			
Available patients	61 (62.9)	2 (33.3)	2 (33.3)
t(11;14)	12 (19.7)	0 (0.0)	0 (0.0)
t(4;14)	8 (13.1)	0 (0.0)	0 (0.0)
t(14;16)	0 (0.0)	0 (0.0)	0 (0.0)
Deletion 17p	1 (1.6)	0 (0.0)	0 (0.0)
International Staging System, *n* (%)			
Available patients	97 (100)	4 (66.7)	6 (100)
Stage I	15 (15.5)	1 (25.0)	1 (16.7)
Stage II	45 (46.4)	1 (25.0)	3 (50.0)
Stage III	37 (38.1)	2 (50.0)	2 (33.3)

CR, complete response; PD, progressive disease; FISH, fluorescence in situ hybridization. * FISH was not performed in these patients, and t(11;14) was detected by chromosome analysis. ** Data at diagnosis are shown in the table except ages of patients in CR and PD, which were determined at study enrollment.

**Table 2 jcm-10-04717-t002:** Median and range of measured values of myeloid-derived suppressor cells and correlation with prognostic markers in 97 newly diagnosed patients with plasma cell myeloma.

MDSCs	Median (Range)	Age	BM Cellularity	BM Plasma Cells	Ca	Cr	LD	β2M	Alb	Hb
		Spearman’s *ρ*								
PB-mMDSCs, %	0.1 (0–1.7)	−0.070	0.142	0.074	0.122	0.430 **	0.441 **	0.353 **	−0.117	−0.316 **
PB-gMDSCs-N, %	22.8 (1.1–56.1)	0.089	−0.094	−0.181	0.121	0.234 *	0.265 *	0.168	0.034	−0.145
Number of PB-mMDSCs, 10^6^/L	7.7 (0–394.4)	−0.081	0.134	0.039	0.161	0.430 **	0.438 **	0.355 **	−0.084	−0.239 *
Number of PB-gMDSCs-N, 10^6^/L	1128 (62–8491)	0.018	−0.062	−0.172	0.163	0.279 **	0.293 **	0.195	0.062	−0.059
BM-mMDSCs, %	0.2 (0–1.0)	0.037	0.040	−0.120	0.064	0.265 **	0.250*	0.217 *	−0.028	−0.159
BM-gMDSCs-N, %	24.9 (0.3–59.9)	0.168	−0.265 **	−0.303 **	0.035	0.165	0.177	0.040	0.064	−0.075

PB-mMDSCs, peripheral blood monocytic myeloid-derived suppressor cells; PB-gMDSCs-N, peripheral blood granulocytic myeloid-derived suppressor cells with neutrophils; BM-mMDSCs, bone marrow monocytic myeloid-derived suppressor cells; BM-gMDSCs-N, bone marrow granulocytic myeloid-derived suppressor cells with neutrophils; BM; bone marrow; Ca, calcium; Cr, creatinine; LD, lactate dehydrogenase; β2M, β2-microglobulin; Alb, albumin; Hb, hemoglobin. * *p* value < 0.05, ** *p* value < 0.01.

**Table 3 jcm-10-04717-t003:** Comparison of laboratory and clinical findings of two groups divided by the level of monocytic myeloid-derived suppressor cells of peripheral blood in 97 patients newly diagnosed with plasma cell myeloma.

Parameters	PB-mMDSCs < 0.3%	PB-mMDSCs ≥ 0.3%	*p* Value
Patients, *n*	69	28	
PB-mMDSCs, %	0.1 (0–0.2) *	0.4 (0.3–1.7)	<0.001
PB-gMDSCs-N, %	19.7 (3.4–48.4)	32.3 (1.1–56.1)	<0.001
Number of PB-mMDSCs, 10^6^/L	4.8 (0–21.4)	26.2 (6.6–394.4)	<0.001
Number of PB-gMDSCs-N, 10^6^/L	990 (66–4490)	2136 (62–8491)	<0.001
BM-mMDSCs, %	0.1 (0–0.8)	0.4 (0.1–1.0)	<0.001
BM-gMDSCs-N, %	22.3 (0.3–58.6)	26.0 (0.7–59.9)	0.442
Age, y	68 (46–87)	68.5 (37–83)	0.793
Age ≥ 65 y, *n* (%)	43 (62.3)	16 (57.1)	0.636
Calcium, mg/dL	8.9 (7.2–17.2)	9.0 (7.7–12.5)	0.786
Creatinine, mg/dL	0.85 (0.41–6.46)	1.43 (0.46–8.2)	<0.001
LD, IU/L	180 (82–526)	227.5 (125–594)	0.001
β2M, μg/mL	3.7 (1.4–28.2)	7.1 (2.5–75.2)	<0.001
Albumin, g/dL	3.3 (1.6–4.4)	2.5 (1.4–4.6)	0.018
M-protein, g/dL	2.1 (0–10)	2.0 (0–5.4)	0.981
WBC count, 10^9^/L	4.9 (1.6–12.2)	6.1 (2.2–23.2)	0.007
Hemoglobin, g/dL	9.9 (5.2–15.4)	8.6 (4.4–11.0)	<0.001
Platelet count, 10^9^/L	176 (54–433)	175 (40–510)	0.484
Lytic bone lesion present, *n* (%)	53 (75.7)	20 (74.1)	0.970
BM plasma cells, %	23.2 (2.4–98.4)	32.0 (10–85.4)	0.176
BM cellularity, %	40 (15–100)	58 (10–100)	0.093
ISS stage I, *n* (%)	15 (21.7)	0 (0.0)	-
ISS stage III, *n* (%)	19 (27.5)	18 (64.3)	0.001
High risk cytogenetics, *n*/*n* (%) **	27/56 (48.2)	13/18 (72.2)	0.075
Mortality, *n* (%)	33 (47.8)	23 (82.1)	0.002

PB-mMDSCs, peripheral blood monocytic myeloid-derived suppressor cells; PB-gMDSCs-N, peripheral blood granulocytic myeloid-derived suppressor cells with neutrophils; BM-mMDSCs, bone marrow monocytic myeloid-derived suppressor cells; BM-gMDSCs-N, bone marrow granulocytic myeloid-derived suppressor cells with neutrophils; LD, lactate dehydrogenase; β2M, β2-microglobulin; WBC, white blood cell; ISS, International Staging System. * Median (range); ** High-risk cytogenetic features are defined as the presence of t(4;14), t(14;16), deletion 17p, and hypodiploidy with structural aberrations in conventional cytogenetics or fluorescence in situ hybridization and deletion 13q in conventional cytogenetics (monosomy 13).

**Table 4 jcm-10-04717-t004:** Cox regression analysis in 81 patients newly diagnosed with plasma cell myeloma.

Prognostic Marker	Univariate			Multivariate		
	HR	95% CI	*p* Value	HR	95% CI	*p* Value
Age ≥ 65 y	1.218	0.661–2.246	0.527	0.864	0.382–1.953	0.725
ISS stage			0.014			0.299
III vs. I	4.182	1.436–12.183	0.009	2.938	0.746–11.568	0.123
III vs. II	1.853	1.001–3.431	0.050	1.480	0.593–3.692	0.400
High risk cytogenetics *	2.132	1.029–4.417	0.042	1.151	0.471–2.814	0.757
PB-mMDSCs ≥ 0.3%	3.610	1.957–6.659	<0.001	2.840	1.049–7.691	0.040
PB-mMDSCs, %	5.896	2.606–13.338	<0.001			
PB-gMDSCs-N, %	1.012	0.987–1.038	0.336			
Number of PB-mMDSCs, 10^6^/L	1.009	1.005–1.014	<0.001			
Number of PB-gMDSCs-N, 10^6^/L	1.000	1.000–1.000	0.114			
BM-mMDSCs, %	5.291	1.597–17.534	0.006			
BM-gMDSCs-N, %	0.982	0.960–1.005	0.131			
Calcium ≥ 10.0 mg/dL	0.967	0.382–2.451	0.944			
Creatinine ≥ 2.0 mg/dL	1.711	0.897–3.266	0.103			
LD ≥ 250 IU/L	2.655	1.404–5.021	0.003	1.439	0.556–3.729	0.453
β2M ≥ 5.5 μg/mL	2.213	1.221–4.009	0.009			
Albumin < 3.5 g/dL	2.071	1.044–4.105	0.037			
Hemoglobin < 8.5 g/L	1.761	0.933–3.322	0.081	1.140	0.397–3.272	0.808
BM plasma cells, %	1.014	1.002–1.026	0.017			
BM cellularity, %	1.015	1.004–1.027	0.006			

HR, hazard ratio; CI, confidence interval; ISS, International Staging System; PB-mMDSCs, peripheral blood monocytic myeloid-derived suppressor cells; PB-gMDSCs-N, peripheral blood granulocytic myeloid-derived suppressor cells with neutrophils; BM-mMDSCs, bone marrow monocytic myeloid-derived suppressor cells; BM-gMDSCs-N, bone marrow granulocytic myeloid-derived suppressor cells with neutrophils; LD, lactate dehydrogenase; β2M, β2-microglobulin. * High risk cytogenetics is defined as the presence of t(4;14), t(14;16), deletion 17p, and hypodiploidy with structural aberrations in conventional cytogenetics or fluorescence in situ hybridization and deletion 13q in conventional cytogenetics (monosomy 13).

## Data Availability

The datasets used and analyzed during the current study are available from the corresponding author on reasonable request.

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
