# Peer review of "Increased Monocytic Myeloid-Derived Suppressor Cells in Whole Blood Predict Poor Prognosis in Patients with Plasma Cell Myeloma"

_jcm, 2021, doi:10.3390/jcm10204717_

Round 1
Reviewer 1 Report
Thanks for the opportunity to review the work titled Increased monocytic myeloid-derived suppressor cells in whole blood predict poor prognosis in patients with plasma cell myeloma, which demonstrate author’s approach to the quantitation of major MDSC and their impact on myeloma patient risk stratification. The work is well written, and the conclusions are well supported by the data. The results are relevant for the field.
Nonetheless, I have the following comments:
-Major comments:
- My main concern with the present text is that because of the choice of type samples to be investigated (WB and BM) and the gating strategy applied, the cell population identified and analysed as gMDSC also contains a high proportion of mature neutrophils. The results, conclusions, and lack of a relevant prognostic impact of this population could be a consequence of the methodology. Authors correctly acknowledged this fact, supported by the absence of a positive identification marker, nonetheless authors chose to keep labelling this population as gMDSC throughout the text. This might be misleading for the readership, and I recommend authors to label this population differently, stating from the beginning that they include not only gMDSC, but also, provably a higher proportion, of mature neuthrophils.
- Inclusion in the text, or probably as supplemental material, of scatter plots used to illustrate correlations and ROC analysis will enrich the text and help the readership to have a more complete view of the results.
- Since there’s no significant difference for the gMDSC, ROC analysis may be not informative, as the prognostic impact shows latter.
-Minor comments:
- In the last paragraph of the introduction, line 54, authors describe mMDSC phenotype as CD14 negative. Please correct.
- In the statistical analysis section, please introduce the symbol used for the Spearman’s rank correlation (“ρ”), since it may be latter confused with “p” value statements.
Author Response
Reviewer: 1
The modifications based on reviewer 1’s comments are colored in yellow in the revised manuscript.
Major comments
- My main concern with the present text is that because of the choice of type samples to be investigated (WB and BM) and the gating strategy applied, the cell population identified and analysed as gMDSC also contains a high proportion of mature neutrophils. The results, conclusions, and lack of a relevant prognostic impact of this population could be a consequence of the methodology. Authors correctly acknowledged this fact, supported by the absence of a positive identification marker, nonetheless authors chose to keep labelling this population as gMDSC throughout the text. This might be misleading for the readership, and I recommend authors to label this population differently, stating from the beginning that they include not only gMDSC, but also, provably a higher proportion, of mature neuthrophils.
We agree with your recommendations. Thus we have changed the labeling to gMDSCs-N (granulocytic myeloid-derived suppressor cells with neutrophils) in the revised manuscript, and also described the cell population in the introduction section.
2. Inclusion in the text, or probably as supplemental material, of scatter plots used to illustrate correlations and ROC analysis will enrich the text and help the readership to have a more complete view of the results.
We have added scatter plots of correlations (supplemental figure 1) and ROC curves of mMDSCs (supplemental figure 2).
- Since there’s no significant difference for the gMDSC, ROC analysis may be not informative, as the prognostic impact shows latter.
We only have added ROC curves of mMDSCs in supplemental figure 2.
Minor comments
1. In the last paragraph of the introduction, line 54, authors describe mMDSC phenotype as CD14 negative. Please correct.
We have corrected CD14- to CD14+ .
- In the statistical analysis section, please introduce the symbol used for the Spearman’s rank correlation (“ρ”), since it may be latter confused with “p” value statements.
We have added (ρ) in the statistical analysis section.
“Spearman’s rank correlation coefficient (ρ) and…”

Reviewer 2 Report
The authors applied a 5-colour flow cytometry strategy to evaluate prognostic impact of monocytic and granulocytic-derived suprresor cells in blood and bone marrow samples within multiple myeloma (MM) patients.
Their study includes analysis of 97 newly diagnosed MM patients. However, only 81 of them were elegible for therapy. Could the authors explain why 16 newly diagnosed MM patients did not treated?
On the another hands, authors refered that 72 from 97 patients were previosly enrolled on published circulating tumour plasma cells (CTCs) study. Did the authors analyse some relationship between presence of CTCs in blood and number of g- or m-MDSCs?
From the statistical point of view, scientific soundness could be improve by analysing median and range values instead of mean and SD. Could the authors recalculated statistical data taking into consideration non-parametric statistical tests?
Figure 1 shows gating strategy of myeloid MDSCs in bone marrow aspirates. Could be possible display a case with >0.3% of mMDSCs and specify percentage of each cell population?
Regarding to Cox regression analysis, Did the authors evaluate data for progression free survival rates? It could improve the conclusion mark of the study.
Author Response
Reviewer: 2
The modifications based on reviewer 2’s comments are colored in mint in the revised manuscript.
- Their study includes analysis of 97 newly diagnosed MM patients. However, only 81 of them were elegible for therapy. Could the authors explain why 16 newly diagnosed MM patients did not treated?
Majority of the patients who did not start myeloma therapy at the time of diagnosis was whose disease status was decided as asymptomatic myeloma by their physicians. Other five patients had medical problems to delay myeloma therapy, such as infection or underlying diseases, they died before starting myeloma therapy.
On the another hands, authors refered that 72 from 97 patients were previosly enrolled on published circulating tumour plasma cells (CTCs) study. Did the authors analyse some relationship between presence of CTCs in blood and number of g- or m-MDSCs?
CTCs of 72 patients showed inverse correlation with BM-gMDSCs (%) level in Spearman’s correlation analysis (ρ = -0.267, p = 0.024) and did not significantly correlated with other MDSCs levels.
From the statistical point of view, scientific soundness could be improve by analysing median and range values instead of mean and SD. Could the authors recalculated statistical data taking into consideration non-parametric statistical tests?
We have recalculated and changed to median and range values in the revised manuscript.
Figure 1 shows gating strategy of myeloid MDSCs in bone marrow aspirates. Could be possible display a case with >0.3% of mMDSCs and specify percentage of each cell population?
We have changed the figure to the case with >0.3% of mMDSCs, and described the percentage of MDSC populations within revised figure 1.
Regarding to Cox regression analysis, Did the authors evaluate data for progression free survival rates? It could improve the conclusion mark of the study.
We agree with your comment. We performed the progression free survival analysis, but MDSCs levels were not significantly associated with progression free survival. We think the exact progression time may not be properly recorded through the long follow-up duration of this study.

Round 2
Reviewer 2 Report
The present article added impact on the prognostic value of blood Monocytic Myeloid-derived Suppressor Cells in newly diagnosed patients. The systematic evaluation of the above cell populations could further contribute to understand the biology of the disease.